# Become Competent within One Day in Generating Boxplots and Violin Plots for a Novice without Prior R Experience

**DOI:** 10.3390/mps3040064

**Published:** 2020-09-23

**Authors:** Kejin Hu

**Affiliations:** Department of Biochemistry and Molecular Genetics, School of Medicine, University of Alabama at Birmingham, Birmingham, AL 35294, USA; kejinhu@uab.edu

**Keywords:** boxplot, box plot, violin plot, *vioplot*, RStudio, R, tutorial, protocol, data visualization, data graphics

## Abstract

The boxplot is a powerful visualization tool of sampled continuous data sets because of its rich information delivered, compact size, and effective visual expression. The advantage of boxplots is not widely appreciated. Many top journals suggest that boxplots should be used in place of bar charts, but have been wrongly replaced by bar charts. One technical barrier to the usage of boxplots in reporting quantitative data is that bench scientists are not competent in generating boxplots, and are afraid of R, a programming tool. This tutorial provides an effective training material in that even a novice without prior R experience can become competent, within one day, in generating professional boxplots. The available R scripts for boxplots are very limited in scope and are aimed at specialists, and the bench scientists have difficulty in following these scripts. This tutorial provides extensive step-by-step R scripts and instructions, as well as 29 illustrations for customizing every detail of the boxplot structures. Basic R commands and concepts are introduced for users without prior R experiences, which can be skipped by audiences with R knowledge. Violin plots are the enhanced version of boxplots, and therefore, this tutorial also provides a brief introduction and usage of the R package *vioplot* with one additional illustration. While the protocol is prepared for the newbies and trainees it will be a handy tool for infrequent users, and may benefit the experienced users as well since it provides scripts for customizing every detail of boxplots.

## 1. Introduction

Boxplots are commonly used to visualize continuous data in all areas including scientific reports of experimental data [1,2,3,4]. There are repeated and continuous calls from the top journals and experts that boxplots should be used in place of bar charts and mean-and-error plots [4,5,6,7,8]. Bar plots are misleading due to the use of a baseline, usually zero, and their inability to present the distribution of a data set [5,9,10,11]. The overwhelming use of bar charts in biological and biomedical sciences is one of the factors for irreproducibility of experiments [12,13]. Bar plots provide only one statistical summary of a data set, the mean, while boxplots provide much more information using a small space, including the minimum, first quartile, median, third quartile, maximum, outlier, spread and skewness of the data set, as well as confidence interval of the median [4,9]. Mean-and-error plots can provide one more characteristic of a dataset in addition to the mean and the error bar, but the visualization effect is very weak compared to boxplots. In their survey of the articles published in *Nature Methods* in 2013, Krzywinski and Altman found that 100 figures were prepared with bar plots and only 20 were prepared with boxplots [4]. There are several reasons for this undue scarce usage of the powerful boxplots in scientific articles. First, the advantages of boxplots are not well advertised. Second, boxplots are more difficult to prepare than bar charts. Third, tools for generation of boxplots are lacking. For example, Excel includes boxplot templates for the latest version only (version after 2016). Most importantly, scientists are afraid of learning programming tools such as R to prepare graphs. In fact, one does not need to have a degree in computer science or statistics to use R just as one does not need to be an engineer to drive a car. One does not need a long training process to prepare boxplots using R as well. This tutorial will help a novice even without prior R experience to become competent within one day in generating professional boxplots. My experience is that biological scientists without prior R experience can become competent within one day in generating RNA-seq heat maps using R, which seems more challenging than generating box plots [14].

This tutorial includes four major parts: (1) An introduction to boxplots; (2) Introduction to basic R concepts and commands; (3) A tutorial of boxplotting in the RStudio platform; (4) A brief introduction of the R package *vioplot* to generate violin plots.

## 2. A Brief Introduction to Boxplots

Box plots are visualization tools for sampled data sets with >5 observations. This idea was first proposed by Mary E. H. Spear, and later popularized by Tukey. Boxplots are also called box-whisker plots because of the their appearance. There are five major components in a boxplot representing five characteristics of a data set, the minimum, first quartile, median, third quartile, and maximum (Figure 1A). On account of this, boxplots are also known as five-number summaries of a data set although other information (outliers, confidence interval, and skewness) may be visualized in the box plots. The lower side of the box is the first quartile (Q1) or 25th percentile while the upper side of the box is called the third quartile (Q3), or 75th percentile. The line that crosses the box is the median. The lines extending from the Q1 and Q3 are whiskers, the ends of which represent the minimum and maximum, respectively. The median value represents the observation that divides the entire data set into two halves when the data set is ordered from the smallest to the largest. The Q1 is the median of the lower half of the data set while Q3 is the median of the upper half of the data set. The box covers the central half of the data set. The value Q3 − Q1 is the interquartile range (IQR = Q3 − Q1). Of note, the minimum is not necessarily the smallest value and the maximum is not necessarily the largest one in the data set. To decide the minimum and the maximum, we have to calculate the locations of fence, which can be calculated by Q3 + 1.5 × IQR for the upper fence and Q1 − 1.5 × IQR for the lower fence. The minimum is the smallest value above the lower fence, while the maximum is the largest value below the upper fence. Any value outside the two fences is an outlier. Outliers are indicated by asterisks or circles. Please note that fences are not drawn in the box plots, and they are used to decide outliers, minimum and maximum only. Box plots may have an extra element of a notch. The notch represents the 95% confidence interval for the median. It is calculated using the formula: CI = median ± 1.58 × IQR/√n in the R *boxplot()* function. The calculations for boxplot components vary among software, and the R *boxplot()* function uses the conventional Tukey method. In the Tukey boxplot, Q1 and Q3 are called the lower and upper hinge. An enhanced version of boxplots is violin plots, in which data density is additionally visualized as smoothed histograms along the data points (Figure 1B) [15].

## 3. Materials

A computer, R software, RStudio software, and data for the generation of the boxplot are required (Appendix A). The data used in this tutorial are the RNA-seq normalized read counts for the 310 human transcription factors that are differentially expressed between human embryonic stem cells (ESCs) and fibroblasts based on analyses in the author’s lab [16], and more information is given in the introduction to the subsection for generating boxplots below. The data are available as a Appendix A, and also available from the author upon request. Both R and RStudio have to be installed. Frequently asked questions (FAQ) for installation of R and RStudio can be found from the references cited [17,18].

## 4. Become Familiar with Some Basic Concepts, Commands, and Functions of R

R is not as difficult as you think. The R commands are intuitive. If u hav no prblem in reding this sentns, u cn use R w/o dificlty. With the launch of the more user-friendly RStudio in 2010, R is no longer the exclusive tool for data scientists, and has been becoming more accessible by users in all areas who have data to be analyzed. You just need to know some very basic concepts and commands to generate professional boxplots. This section will familiarize you with such basic concepts and commands before I will guide you to generate boxplots using the generic function of R, *boxplot()*. For emphases, I will highlight any R function, command, or operator by italicization in the text as you can see in the last sentence for the R function of *boxplot()*. More specific commands/functions will be introduced in the next section when they will be used to implement the *boxplot()* function. For the best results, please practice on a computer what the tutorial instructs. Do not just read this tutorial. Those with R knowledge may skip this section. To generate violin plots, the R pacakage *vioplot* should be installed separately onto RStudio from the Comphrehensive R Archive Network (CRAN) as instructed in the tutorial below. 

Download R, and then RStudio. Please note that you have to download both software because RStudio needs R to work. Open your RStudio just as you open any application on your computer. You will see an R window with four panes: Source Editor/Data Viewer, Console, Environment/History, and Plots/Files/Packages/Help. Think of the RStudio console as an electronic calculator, and you can conduct similar calculations as you do with a calculator and immediately see the results on the console pane. In the R Console pane, you will see a prompt sign >. This > means you can input R commands here. Type down any operation with the arithmetic operators *+*, *−*, *, /, and ^. In this tutorial, the commands you will input to practice are indicated by blue color. Text after # following the commands/scripts is a description of or comments about the R commands/functions. I will not give the results/outputs in the tutorial because of space limit but I may describe the outcome sometime after # following the script.

5 + 10    # You have to hit the Enter key to tell the computer to execute your commands. I omit this step in this tutorial to save space. You will see the result of the above calculation, 15, immediately after you press the Enter key.(16 - 6)/5  # you will see the result, 2, immediately.5^2     # The result, 25, appears underneath the command line.

You can conduct calculations using the trigonometric, or logarithm functions *cos(x)*, *sin(x)*, *tan(x)*, *log(x, base)*, *log2(x)*, and *log10(x)*.


log(8, 2)
log2(8)   # The above two calculations give you the same result, 3.

R absolutely does much more than a calculator does. One key concept is that you can store any result of the above calculations as an **object** that can be used in later calculations and manipulations. You need an **assignment operator** to store the result as an object. In R, the widely used **assignment operator** is the combination of an arrow and a minus sign <-, which should stick to each other without any space between the two. Please note that the arrow of the assignment operator should point towards the object and the object can be before or after the assignment operator. Please type the blue script below:x <- 5 +10     # It assigns the result of 5 + 10 to the object x.5 + 10 -> x    # This functions the same as the above command.y <- (16 - 6)/5  # It assigns the result of (16 - 6)/5 to the object y.z <- 5^2c <- log2(8)

You cannot see the results immediately for the above calculations even though you have hit the Enter key. The results are saved on the active memory of R as **objects** x, y, z, and c. Try typing these object names and you will see the results on the screen.


x

y

z

c


Typing the above objects is an implicit use of the **function**
*print()*:print(x)  # You see the same result as you type x.print(y)print(x)print(c)

Object names can be a combination of letters, numbers, dots, and underscores, but cannot start with a number or an underscore. Objects in R can be anything: numbers, character strings, results, vectors, arrays, matrix, data frames, or R functions. You can assign several objects in one line, each separated by a semicolon. Please note that the R **character string** should be enclosed with quotation marks.

name <- “Kejin Hu”; Kejin_Hu <- “author”; x1 <- 5*10/2; n2 <- 4k; me <- “reader” # “Kejin Hu”, “author”, and “reader” are R character strings.

Please note that object names are case sensitive. Please type:Kejin_hu # It results in a warning sign: *Error, object “Kejin_hu” not found*. This is because the lowercase h was used in the object name.Kejin_Hu  # It returns “author” on the screen.Me    # Because capital M is used, it results in a warning sign: *Error, object “Me” not found*.me    # It returns “reader” on the screen, which is the content of the object *me*.

Please also note that the object name should match exactly. For example, type

Kejin.Hu # It results in a warning: *Error: object ‘Kejin.Hu’ not found.* An underscore should be used here. You can try typing Kejin_Hu, and you will get a return of “author”.

One very useful function is *help()*, or ?.


help(ls)
?ls  # Both of the help functions will bring up the online built-in document about the list function, *ls()*.

A help window will pop up, which contains information about how to use the function in question. A help document generally includes sections of Description, Usage, Arguments, Details, References, See Also, and Examples. The help command can be used as *help(“print”)* or *help(print)*, but if you want to retrieve the help page for non-conventional characters (+, −, *, /, ^, ?), the character should be enclosed with quotation marks.

help(?)      # This results in a warning: *Error, unexpected ‘)’ in “help(?)”*.help(“?”)  # This command will bring about the help page for the function of *?*.

An R object has different **modes** including numeric, character, complex, and logical. The mode of the object can be found using the function of *mode()*.

mode(name) # It tells you the object *name* is a character one.mode(x1)   # It tells you that x1 is a numeric object.mode(n2)   # It tells you that n2 is a complex object.

One basic data structure of R is the **vector**. You can generate a vector object using the combine function of *c()* and assign the resulting vector to a specific name:temperature <- c(29, 30, 32, 31, 35, 37, 38)wind <- c(7, 5, 4, 10, 8, 6, 9)city <- c(“Madison, WI”, “Detroit, MI”, “Chicago, IL”, “Rochester, NY”, “Birmingham, AL”, “Miami, FL”, “Austin, TX”)   # Please note that a character string has to be enclosed with quotation marks.

One very useful function is repeat *rep()*. For example:region <- c(rep(“north”, 4), rep(“south”, 3)) # This generates a character vector of 7 elements with two different character strings. The code *rep(“north”, 4)* instructs R to repeat the “north” character string 4 times.mode(temperature)  # It tells you that temperature is a numeric vector.mode(wind)mode(city)  # It tells you that city is a character vector.

Another intrinsic attribute of an object is its **length**. The length of any object can be found using the function of *length()*.

length(temperature) # It tells you the length of the temperature vector is 7.

You can combine the vectors *city*, *region*, *temperature*, and *wind* to form a table. Because this table contains both character and numeric values, the R term for this table is **data frame**. We use the function *data.frame()* to generate this table (data frame), and assign a data frame name of weather to it.


weather <- data.frame(city, region, temperature, wind)


Type the data frame name and see what will appear on the screen:weather  # As you can see, this data frame has four columns. Two columns are character strings, and the other two are numeric values.

We can extract column 3 and 4 to form a **numeric matrix** using the extract operator [ ]:weather_data <- weather[, c(3, 4)] # Here, the first comma indicates that all rows are extracted and kept in the new numeric matrix of weather_data; c(3, 4) means column 3 and 4 are taken and kept in the new matrix. Alternatively, you can subtract columns 1 and 2 to obtain the same matrix: weather_data <- weather[, c(−1, −2)]  # Here, - means dropping columns 1 and 2. The first comma denotes that all rows will remain. 

Type the **matrix** name weather_data to see what will appear on the screen.


weather_data


We can conduct calculations on the entire **numeric matrix**, for example:weather_data1 <- log2(weather_data) # After running this script, type weather_data1 to see the result.weather_data2 <- weather_data*2  # After running this script, type weather_data2 to see the result.

Another widely used operator is the colon sequence operator: The following script uses the colon sequence operator: to extract the weather data for the four northern cities, i.e., the first four rows of the data, to form a shortened data frame, north_city:north_city <- weather[1 :4 ,]   # [ ] means to extract elements from the data frame of weather; 1:4 means row 1 to 4 will be extracted; the comma in the column position indicates that all columns will be kept in the new data frame.

You can generate a numeric vector of 1 to 10 using the colon operator:number <- c(1:10)number # The result on the screen will help you understand how c(1:10) works.

**Retrieve and edit previous commands**: Frequently, you want to recall the previously used commands in R. This can be done simply by using the up- and down-arrow keys. Hitting the uparrow key one time will bring about the last command you just executed; keep hitting the uparrow key and the previously used commands will appear in the reverse order of execution. The downarrow key will do the reverse. When the command of your interest appears, you can re-execute it by hitting the Enter key, or edit the scripts and execute the new script. You can also bring about the list of the commands you just used in a popup window, and select the command/script of your interest from the list. In RStudio, this can be done by a combination of ctrl/uparrow on PC or Command/uparrow on MAC. Practice the recall operations described above.

All objects are saved on the active memory of R. You can use the *ls()* function to list all the active objects of your R session.

ls()  # You will see all of the objects you practiced above: “x”, “y”, “z”, “c”, “name”, “Kejin_Hu”, “x1”, “n2”, “me”, “temperature”, “wind”, “region”, “weather”, “weather_data”, “weather_data1”, “weather_data2”, “norther_city”, “number”.

You can remove any object from the R active memory using the remove function *rm()*:rm(x)  # This removes the object x. You can type x, and you will get a warning as: *Error: object ‘x’ not found*.

You can remove as many objects as you want by listing the object names in the *rm()* function:rm(y, z, c, name, Kejin_Hu)  # After running this script, try to type each of these object names and see the outcomes.

To quit RStudio, run:q(“yes”)

## 5. Generating Boxplots Using the Generic R Function *boxplot()*

Generating boxplots using R is not more difficult than conducting a task in Excel if you have a good tutorial like this one. We do not need to know all the R commands to generate a boxplot. This is also true for Excel. It is said that an ordinary user uses less than 5% of the available Excel functions. This section instructs readers on how to generate boxplots in RStudio. When needed, I continue to introduce additional functions, commands and operators required for generating and enhancing boxplots. Please note that this tutorial provides more techniques than needed to conduct routine boxplotting, but there may be other scripts to achieve the same goals. These are also incomplete descriptions of the function of *boxplot()* and the associated R functions. I suggest that you type the scripts in this tutorial because it may not work if you copy the scripts from the PDF file of this article and paste them into the RStudio because of cryptic coding of letters/symbols within the document.

Let’s start with an Excel file of RNA-seq data (Table file boxplot_TF_ESC_hi.xlsx). This Excel table contains 310 human genes of transcription factors that are expressed significantly higher in human embryonic stem cells (ESCs) than in human fibroblasts by at least 2-fold at the significant levels of q < 0.01. Three ESC and four fibroblast samples were sequenced. Boxplots can readily demonstrate these differences for the large sets of genes across seven RNA-seq samples.

## 6. Import Experimental Data Into R, Prepare and Manipulate the Numeric Matrix

**Saving an Excel file as a comma-separated values (CSV) file**. First, we save the RNA-seq Excel file in CSV format so that we can have RStudio to read the file using the *read.csv()* function. It is recommended that you save the original .xlsx or .xls file as Unicode Tranformation Format – 8-bit (UTF-8) CSV since the author sometimes experienced problems with other CSV formats. Please save the provided demo Excel file as boxplot_TF_ESC_hi.csv in the boxplot folder of users/myfolder/boxplot since I wrote this tutorial with such a file and path name.

**Finding out and setting the working directory**. You can now open RStudio and find out the working directory of R using the command *getwd().*

getwd() # The working directory will be printed out on the screen within the R console.

Make sure that the R working directory is the same one where your RNA-seq file for boxplotting is saved. Suppose it is C:/users/myfolder/boxplot. If not, set your *boxplot* folder as the working directory using the function of *setwd()*.

setwd(“C:/users/myfolder/boxplot”)  # Now, the hard drive folder containing the RNA-seq result file becomes the working directory of R. You can check it using *getwd()*. Please note that you have to enclose the path with quotation marks when using the *setwd()* function.

In RStutio, there are two simpler ways to set the target folder as the working directory. First, from the main menu of RStudio, expand the “Session” pulldown menu just as you do for any Windows menu, then, go to the “Set Working Directory” submenu, and click on “Choose Directory”. This will bring about the dialogue window, which gives you access to your computer directory. Choose the directory you want to use as the working directory (containing your RNA-seq file). Click “Open”. You can see that the path appears in the console pane. The directories in the Files window of the lower right pane also change correspondingly.

You can also change the working directory from the lower right pane. Click the Files menu on the Files/Plots/Packages/Help/Viewer pane → click myfolder → select the boxplot folder → click the “More” pulldown menu → in the pulldown menu, select “Set As Working Directory”. In the R console pane, you will see that setwd(“users/myfolder/boxplot”) was executed and the boxplot folder became the working directory.

**Read data file into R**. Read the file from the computer hard drive into the R active memory, and assign the data frame an object name of *esc2xhigh* using the function *read.csv()*:esc2xhigh <- read.csv(“boxplot_TF_ESC_hi.csv”) # Please do not forget the file extension .csv. Otherwise, the *read.csv* function cannot be executed and you will see a warning.

In the Environment window on the upper left, you now see the data frame name of *esc2xhigh* with the number of rows (objects, 310) and number of columns (variables, 24). Click on the data frame name of *esc2xhigh*, and you will see the RNA-seq data on the Data Viewer pane in the upper left quadrant.

**Generating numeric matrix from the numeric parts of a data frame**. Out of the 24 columns of the table, we need the seven columns of normalized read counts only for generating boxplots, i.e., column 14 to 20. For boxplot preparation, we generate a numeric matrix with the seven columns of read counts. To do this, we use the index operator *[ ]*, and assign the matrix a name of esc2xhigh1.

esc2xhigh1 <- esc2xhigh[, 14:20] # The comma “,” here instructs that all the 310 rows will be kept in the new matrix. The “14:20” here denotes that column 14 to 20 are extracted from the data frame and put into the new matrix of esc2xhigh1. Note that 14:20 can be replaced with *c(14, 15, 16, 17, 18, 19, 20)*, that is, *esc2xhigh1 <- esc2xhigh[, c(14,15, 16, 17, 18, 19, 20)]*. Now, click on the matrix name esc2xhigh1 in the Environment pane, and you can see the matrix in the Data Viewer pane (upper left).

## 7. Briefly Examine the Matrix

You can see the first n rows of your new matrix in the R console pane using the *head()* function:head(esc2xhigh1, n = 10) # The default number of rows to be displayed in the console is 6. Usually, you type *head(esc2xhigh1)* to see the first 6 rows of your matrix, without specifying the n (using the default value of 6).

You can find out how many rows are in your matrix using the function of *nrow()*:nrow(esc2xhigh1) # This tells you there are 310 rows in the matrix of *esc2xhigh1*. This information is also available in the Environment pane.

You can find out how many columns are in your matrix using the function of *ncol()*:ncol(esc2xhigh1) # This command tells you there are 7 columns in the matrix of *esc2xhigh1*. This information is also available in the Environment pane.

You can list the names of the column of your matrix using *colnames()*:colnames(esc2xhigh1) # It will print out the 7 column names of the matrix *esc2xhigh1*. Similarly, you can list the names of the rows using *rownames()*.

You can find the minimum value in the matrix using the function of *min()*:min(esc2xhigh1) # This will print out the smallest number in the matrix. Similarly, you can find out the largest value in the matrix using *max(esc2xhigh1)*.

You can use the function of *summary()* to see the basic statistical features of your data:summary(esc2xhigh1) # This command prints out the six statistical summaries for each column of your data in the console: minimum, first quartile, median, mean, third quartile, and maximum. Please note that the minimum and maximum revealed by *summary()* are the smallest and largest values, respectively, which are different from the minimum and maximum for drawing box plots (see below).

**Using Data Viewer to examine the data set**. You can see all the data in the Data Viewer pane (upper left pane). You can put the data set of your interest into the Data Viewer by clicking its name in the Global Environment pane. Try to click *esc2xhigh* and check the Data Viewer pane; then, click *esc2xhigh1* and check the Data Viewer pane. Instead of clicking, you can use the command *View()*. Try:View(esc2xhigh)   # Please note that unlike other commands, V in the *View()* command is in uppercase.View(esc2xhigh1) # Check the change in the Data Viewer pane.

You can search your data using the search box (upper right of the Data Viewer pane). For example, when viewing the *esc2xhigh* data frame, inside the search box at the upper right corner (with a magnifying glass) type CBX2 to see what will happen.

You can also sort each column at the column name box. Try to click any column name of the *esc2xhigh1* matrix in the Data Viewer pane, and the click again to see the change of the sorted data.

## 8. Manipulating the Numeric Matrix

You can calculate the entire matrix, for example, by log2:esc2xhigh_log <- log2(esc2xhigh1) # Click *on esc2xhigh_log* in the Environment window, and see the new log2-transformed matrix in the Data Viewer pane. You can also see the first 6 rows of your log2-transformed matrix by running *head(esc2xhigh_log)*, and see the statistical summaries by running *summary(esc2xhigh_log)*. Log2-transformation is the most useful transformation in data visualization, as you will see in the next section.

You can shuffle the columns of a matrix using a numeric vector in combination with the index operator *[ ]*. This operation is useful if you want to re-order the boxplots (see below).

esc2xhigh_re_ordered <- esc2xhigh1[, c(5, 6, 7, 1, 2, 3, 4)] # This script places column 5 to 7 of the matrix esc2xhigh1 in column 1 to 3 of the new matrix esc2xhigh_re_ordered, and column 1 to 4 of the matrix esc2xhigh1 in column 4 to 7 of the new matrix esc2xhigh_re_ordered. You can check the new order of columns using the commands of head(esc2xhigh_re_ordered). Please note that R object names should contain only dots, underscores, numbers and letters, and hyphens are not allowed. Please try esc2xhigh_re-ordered <- esc2xhigh1[, c(5, 6, 7, 1, 2, 3, 4)], and you will find an error message because you use a hypen in the object name.

## 9. Generating the Basic Boxplots


**Make your first boxplot using the raw data by running the function *boxplot()***


The function *boxplot()* is a generic one within the default R package *graphics* without the need of any additional packages to be loaded or downloaded post R installation. *boxplot()* is simple, flexible, yet powerful. You can generate boxplots and enhance every detail of the boxplots, such as the line thickness, style and colors of boxes, whiskers, median lines, staples, and outliers; add and refine figure titles, axis labels, figure legend, and sample names; add axes on all four sides, and conduct cosmetic modifications of axes; change margins, and others. The following texts will walk you through these procedures.

The simplest script for generating boxplots with a matrix such as esc2xhigh1 is simply calling the function *boxplot()* using all the default arguments of *boxplot()*:boxplot(esc2xhigh1) # It results in 7 boxplots in the figure field, each of which is based on one column of read counts, i.e., 310 read counts representing 310 human transcription factors from one RNA-seq experiment for either ESCs or fibroblasts.

You can immediately see boxplots on the Plots pane in the lower right quadrant (Figure 2). However, we cannot see the different components of the boxplots, as illustrated in Figure 1A, except for some outliers (open circles); these boxplots are not informative and do not reveal the differences in gene expression between the two cell types, as we have already known based on fold differences and significance tests.

## 10. Generate High Resolution Boxplots Using the log2-Transformed Data

Use the *summary()* function to see the basic characteristics of the data set related to boxplot configuration:summary(esc2xhigh1)

Now, you see the six features for each treatment of both cell types: minimum (the smallest value), first quartile, median, third quartile, maximum (the largest value), and the mean. You can even see complete and more relevant summaries of the data when you assign the **argument**
*plot* a logical value FALSE or F to execute the *boxplot()* function.

boxplot(esc2xhigh1, plot = FALSE) # Different summaries of the data set are printed out within the Console pane including stats (minimum, 1st quartile, median, 2rd quartile, and maximum of each column), the number of observations of each column (n), the lower and upper extreme of each notch (conf), a list of outliers in each column (out), the group of each outlier (group), and group/column names (names). Please note that the logical value FALSE or F should be in uppercase.

Based on the summaries, the entire ranges of data for each sample/column are very large, but the IQRs are all very small relative to the dataset range. This results in fusion of the different boxplot structural components. One approach to addressing this issue is to transform the RNA-seq read counts by log2 calculation. We can conduct log2-transformation of the entire matrix within the *boxplot()* function.


boxplot(log2(esc2xhigh1))


Now, all the components of the boxplots are easy to see, and the differences of gene expression levels between the two cell types are revealed (Figure 3).

Alternatively, you can transform the matrix esc2xhigh1 by log2, and assign a matrix name to the newly log2-transformed gene expression matrix:esc2xhigh2 <- log2(esc2xhigh1)

Then, you can generate boxplots with the new expression matrix esc2xhigh2:boxplot(esc2xhigh2)

You obtain the same boxplots as Figure 3 generated with the previous script.

## 11. Deal with the -Inf in Your Matrix to be Boxplotted

In both scripts for the log2-transformed data, you see a warning because the outlier -Inf cannot be visualized. We also know that there are other lower outliers in the fibroblast sample 4, although there are -Inf in this sample. You can ignore the -Inf outliers. Alternatively, you can visualize the -Inf outliers as well by replacing -Inf with a value smaller than the smallest value in the data set excluding -Inf. This takes two steps of replacements. First, we find out the largest value in this entire dataset using the *summary()* function:summary(esc2xhigh2)

The summaries of the log2-transformed dataset are printed out. As you can see, the largest value in the dataset is 16.964. Alternatively, you can find the largest values using the sort functions in the Data Viewer pane, but you have to sort 7 times. Unlike the sorting function, *summary()* gives you all the 7 largest values in the 7 columns at one time. If you do not want to know which sample has the largest value, you can quickly find the largest value in the matrix using the *max()* function introduce previously.

Therefore, we can temporally replace -Inf with a large value larger than the largest value in the original matrix, say 20, using the index operator *[ ]*.

esc2xhigh2[esc2xhigh2 == -Inf] <- 20  # The logical operator == means “exactly equal to”. This script replaces all -Inf within the matrix *esc2xhigh2* with the number 20. Please note that the minimum and maximum results based on *boxplot(esc2xhigh2, plot = F)* are different from those obtained using *summary(esc2xhigh2)*, or *min() and max()*. The former function returns the statistical minimum and maximum excluding the outliers. The latter returns the smallest and the largest value as minimum and maximum, respectively. You should use the latter here to find out the largest, as well as the smallest in the next step. Do not use the former script for this purpose. Please note that you have to use the capital I in -Inf.

Then, run *summary()* again for the new matrix to find out the smallest value in the dataset with the -Inf excluded (or using the sorting function, or the *min()* function):summary(esc2xhigh2)

From the summaries, we can see that the smallest value in the dataset is −1.452. Therefore, we can replace the original -Inf with a value smaller than −1.452. Let us replace -Inf, i.e., 20 in the current matrix *esc2xhigh2*, with −2.


escx2xhigh2[esc2xhigh2 == 20] <- −2


Now, execute *summary(esc2xhigh2)* again and you can see that every -Inf has been replaced with −2. In general, replacing -Inf with −3.3 is appropriate since log2(0.1) = −3.3 and log2(0) results in -Inf. However, −2 is better than −3.3 because it will use less space to keep the outliers in the figure.

Generate boxplots again with the new matrix:boxplot(esc2xhigh2)

You see the same boxplots except for more outliers (Figure 4). The new outliers below the lower whisker staples represent -Inf. Please note that the outliers do not affect the configuration of boxplots. Therefore, replacing -Inf with a small number outside of the fence may just make the outliers visible or summarized visually.

Alternatively, we can replace all the zero values, which result in -Inf by log2 transformation, with a very small value in the original matrix of normalized read counts before log2-transformation. For this purpose, we can find out the smallest normalized read counts using the methods described above. The smallest non-zero value in the matrix is 0.3654162. Therefore, replacing zero with 0.2 will be appropriate for the data set with a large data range. For other data set, the readers must decide the best way to deal with the -Inf.

## 12. Re-Order the Boxplots

The order of each box can be specified using the index operator *[ ]*:boxplot(esc2xhigh2[, c(3, 5, 2, 7, 4, 6, 1)]) # As you can see, the boxes can be placed in any order. The first comma indicates that you use data from all rows of each column for boxplotting.

You can combine the index operator [ ] and the sequence colon operator: to order the boxes,

boxplot(esc2xhigh2[, c(5:7, 1:4)]) # This script places the three ESC boxplots before the four fibroblast boxplots (Figure 5). The colon sign means that all columns from the left number to the right number will be extracted and used in the order specified.

Alternatively, you can order the samples/columns in the matrix before you prepare the boxplots:esc2xhigh3 <- esc2xhigh2[, c(3, 5, 2, 7, 4, 6, 1)] # This first comma means that you extract all rows for each column specified.boxplot(esc2xhigh3) # You obtain the same boxplots as those obtained using the code of *boxplot(esc2xhigh2[, c(3, 5, 2, 7, 4, 6, 1)])*.esc2xhigh4 <- esc2xhigh2[, c(5:7, 1:4)] # The first comma means that you extract data from all rows for each column specified.boxplot(esc2xhigh4) # You obtain the same box plots as in Figure 5.

## 13. Enhancements of Boxplots

In the following sections, I will demonstrate how to enhance the presentation of boxplots using additional **arguments** within the function *boxplot()*. For your convenience, I will put the **argument** in discussion the last one and highlight it in boldface. Please note that you can place multiple arguments in any order within the function of *boxplot()*.

We can enhance the visualization of data by modifications of the boxplots using different **arguments** of the *boxplot()* function.

**Add notch to boxplot**. One of the important non-default features of boxplots is notch. We can add notch to each box using the logical value TRUE or T for the argument of *notch* and generate boxplots with notches (Figure 6):boxplot(esc2xhigh2, **notch = TRUE**) # You will see notched boxes now. It is said that if the notches of two samples do not overlap, it is certain that there are significant differences between the two samples. However, overlapping of notches does not mean there is no significant difference. Please note that if the sample size is small or the confidence regions may go out of the bounds of one of the boxes, it may show an error. In this case, do not define the *notch* argument, or set it as FALSE or F.

**Fill the boxes with colors**. You can fill the boxes with a color of your choice, say, gray, using the *col* argument (Figure 7):boxplot(esc2xhigh2, notch = T, **col = “gray”**) # col means colors.

For a list of the colors available in R, use the *colors()* function:colors() # This prints out a list of 657 colors available in R.

**Using different colors to distinguish cell types**. You can mark each box with different colors using a color vector. Let us fill the fibroblast boxes with light blue and the ESC boxes with bisque (Figure 8):boxplot(esc2xhigh2, notch = T, **col = c(“lightblue”, “lightblue”, “lightblue”, “lightblue”, “bisque”, “bisque”, “bisque”)**) # The use of different colors visually groups fibroblasts from ESCs.

We can further distinguish the two cell types by using different colors using the *border*
**argument** (Figure 9):boxplot(esc2xhigh2, notch = T, col = c(rep(“lightblue”, 4), rep(“bisque”, 3)), **border = c(rep(“red”, 4), rep(“violet”, 3))**)  # The *border* argument defines the colors of all boxplot components except for the box body. You can use a color vector to define the colors of each individual box. If a single value of color is used, *border* will assign the same color to all boxplots.

**Horizontal boxplots**. You can present the boxplots horizontally using the logical argument *horizontal* (Figure 10):boxplot(esc2xhigh2, notch = T, col = c(rep(“lightblue”, 4), rep(“bisque”, 3)), border = c(rep(“red”, 4), rep(“violet”, 3)), **horizontal = T**)  # To save space, the colors are defined using the *rep()* function.

**Changing sample names of boxplots**. You can see in Figure 10 that two sample names are missing. This is because the sample names are too long. We can use the *names* argument to specify shorter names (Figure 11).


boxplot(esc2xhigh2, notch = T, col = c(rep(“lightblue”, 4), rep(“bisque”, 3)), border = c(rep(“red”, 4), rep(“violet”, 3)), horizontal = T, **names = c(“F1”, “F2”, “F3”, “F4”, “E1”, “E2”, “E3”)**)


**Spatial separation of the boxplots for the two cell types**. To further visually distinguish the two cell types, you can adjust the location of the boxes using the *at* argument so that boxplots for the two cell types can be spatially separated. Let us move boxes 1 to 4 to the left by 0.3, and boxes 5 to 7 to the right by 0.3, so that the groups of fibroblasts and ESCs can be separated/grouped (Figure 12, compare with Figure 9).

boxplot(esc2xhigh2, notch = T, col = c(rep(“lightblue”, 4), rep(“bisque”, 3)), border = c(rep(“red”, 4), rep(“violet”, 3)), **at = c(1:4 - 0.3, 5:7 + 0.3)**)  # In the numeric vector here, 1:4 - 0.3 means to subtract 0.3 from each of the four numbers 1 to 4; and 5:7 + 0.3 means to add 0.3 to each of the three numbers 5, 6, and 7.

## 14. Modifying Boxes

The arguments of *boxlty*, *boxlwd*, *boxcol*, and *boxfill* can be used to define the line types, line width, line colors, and body colors of the boxes (not of whiskers, staples, and outliers), respectively. Each of these arguments can define individual boxes (using vector values) or all boxes together (use one value). You may notice that *col* can also define the colors of boxes. Please note that if *col* and *boxfill* are both specified, *boxfill* overrides *col*. For values of line types, see Section 15 below. The following is an example script to modify the box components only while the other components of boxplots remain as the default values (Figure 13)


boxplot(esc2xhigh2, **boxlwd = 4, boxlty = c(3, 3, 3, 3, 4, 4, 4), boxcol = c(rep(“magenta”, 4), rep(“brown”, 3)), boxfill = c(rep(“gray”, 4), rep(“lightblue”, 3))**)


## 15. Modifying Whiskers

The color, thickness, and line style of whiskers can be defined using *whiskcol*, *whisklwd*, and *whisklty*. The “whisk” denotes whisker, and *col*, *lwd*, and *lty* mean colors, line width, and line types, respectively. The *lty* values are: 0, blank; 1, solid; 2, dashed; 3, dotted; 4, dotdash; 5, longdash; 6, twodash. The following is a representative script for modifying the whiskers only leaving other parts of boxplots as the default values (Figure 14).


boxplot(esc2xhigh2, **whisklwd = 4, whiskcol = c(rep(“red”, 4), rep(“green”, 3)), whisklty = c(3, 3, 3, 3, 1, 1, 1)**)


## 16. Modifying Outliers

The outlier line width, point character, point size expansion, and color can be defined using *outlwd*, *outpch*, *outcex*, *outcol*, respectively. The “*out*” in these arguments denotes outlier; and *lwd*, *pch*, and *col* mean line width, point character, and color, respectively. Each feature of outliers for each treatment can be defined individually using a vector. If single value is used, all outliers will be defined the same way. The script below is an example (Figure 15), and you can change the values to see the differences.


boxplot(esc2xhigh2, **outlwd = c(3, 3, 3, 3, 2, 2, 2), outpch = c(rep(1, 4), rep(5, 3)), outcex = c(2, 2, 2, 2, 1, 1, 1), outcol = c(rep(“red”, 4), rep(“blue”, 3))**)


There are 25 values for *pch*: 0, square; 1, circle; 2, triangle point up; 3, plus; 4, cross; 5, diamond; 6, triangle point down; 7, square cross; 8, star; 9, diamond plus; 10, circle plus; 11, triangles up and down; 12, square plus; 13, circle cross; 14, square and triangle down; 15, filled square; 16, filled circle; 17, filled triangle point-up; 18, filled diamond; 19, solid circle; 20, bullet (smaller circle); 21, filled circle blue; 22, filled square blue; 23, filled diamond blue; 24, filled triangle point-up blue; 25, filled triangle point down blue.

## 17. Modifying Staples

The arguments *staplelty*, *staplelwd*, *staplecol* and *staplewex* can be used to define line types, line width, line colors, and the line length of the staple part of boxplots, respectively. The values for line types can be found in the Section 15 above. The script below is an example (Figure 16), and you can change the values to see the differences.


boxplot(esc2xhigh2, **staplelwd = 4, staplelty = c(3, 3, 3, 3, 1, 1, 1), staplecol = c(rep(“red”, 4), rep(“cyan”, 3)), staplewex = 1**)


## 18. Modifying the Median Lines

The arguments *medlty*, *medlwd*, and *medcol* can be used to define the style, width and color of the median lines, respectively. The script below is an example (Figure 17), and you can change the values to see the differences.


boxplot(esc2xhigh2, **medlty = c(3, 3, 3, 3, 1, 1,1), medlwd = 6, medcol = c(rep(“red”, 4), rep(“green”, 3))**)


## 19. The Relationship of the *border* Argument and the Specific Color Arguments of Boxplots

We can modify colors of the boxplots using the argument of *border* as well. The *border* argument defines the color of box borders, median lines, whiskers, staples, and outliers. Please note that if *border* is used along with any of *medcol*, *boxcol*, *whiskcol*, *staplecol*, or *outcol*, the component color arguments will override the *border* argument for the corresponding boxplot components. Try the following script, and compare the result (Figure 18) with Figure 9.

boxplot(esc2xhigh2, notch = T, col = c(rep(“lightblue”, 4), rep(“bisque”, 3)), **border = c(rep(“red”, 4), rep(“violet”, 3)), medcol = c(rep(“black”, 4), rep(“brown”, 3))**) # In this script, *medcol* overrides *border*.

## 20. Annotation and Labeling of Box Plots

**Change names of each boxplot**. This can be done in two ways.

You can use the *names* argument within the *boxplot()* function.


boxplot(esc2xhigh2, **names = c(“F1”, “F2”, “F3”, “F4”, “ESC1”, “ESC2”, “ESC3”)**)


Alternatively, you can do it in two steps. First, you change the column names of the matrix using the function *colnames()*:colnames(esc2xhigh2) <- c(“F1”, “F2”, “F3”, “F4”, “E1”, “E2”, “E3”) # After running this script, click the matrix name *esc2xhigh2* in the Environment pane, and check the column names in the Data Viewer pane to see the changes.

Then, you run *boxplot(esc2xhigh2)*, and you can see the differences in x axis labeling. Compare the resulting figure with other figures above. In the following illustrations, we will use the new matrix that has the shorter column names.

**Add figure title**. You can add a title to the boxplot using the argument of *main* (see Figure 19 below):boxplot(esc2xhigh2, notch = T, **main = “Differential expression of transcription factors between fibroblasts and ESCs”**)

**Add axis titles to the x and y axes.** You can annotate the x and y axis using the *xlab* and *ylab* arguments (Figure 19).


boxplot(esc2xhigh2, notch = T, main = “Differential expression of transcription factors between fibroblasts and ESCs”, **xlab = “Sample names”, ylab = “Log2-transformed read counts”)**


**Define orientations of axis labels**. The orientation of the labels for the axes can be defined with the *las* argument as: parallel to the axes (las = 0); horizontal (las = 1); perpendicular to the axes (las = 2); and vertical (las = 3). Parallel is the default. In the following script, the axis labels are vertical (Figure 20). You can try all of the 4 orientations.


boxplot(esc2xhigh2, notch = T, main = “Differential expression of transcription factors between fibroblasts and ESCs”, xlab = “Sample names”, ylab = “Log2-transformed read counts”, **las = 3**)


**Add legend within the figure field**. You can add a color legend to the figure using the *legend()* function. This can be done in two steps. First, draw the boxplot.

boxplot(esc2xhigh2, notch = T, col = c(rep(“lightblue”, 4), rep(“bisque”, 3)), border = c(rep(“red”, 4), rep(“violet”, 3)), las = 3)  # Please note that the orientation of the axis labels are different from the default (compare Figure 20 and other figures above).

Then, you use the *legend()* function to add the legend onto the empty area in the figure field. The location of the legend can be specified. The following script places the legend at the coordinates of (6, 5):**legend(6, 5, c(“fibroblast”, “ESC”), fill = c(“lightblue”, “bisque”), cex = 1.5)**  # You can see the legend at the (6, 5) position (Figure 20). The *cex* argument defines the size of the text; *fill* defines the colors of each of the legend boxes, which should match the colors of the corresponding boxes.

**Change colors of figure annotations and labels**. The colors of the figure title, axis titles, axis labels can be specified using the arguments *col.main*, *col.lab*, and *col.axis*, respectively (Figure of this script is not shown, but the colors of figure title, axis titles, and axis labels are defined the same way as in Figure 23 below):boxplot(esc2xhigh2, notch = T, main = “Differential expression of transcription factors between fibroblasts and ESCs”, xlab = “Sample names”, ylab = “Log2-transformed read counts”, **col.lab = “darkorange”, col.main = “red”, col.axis = “magenta”**)

**Change font sizes of axis titles and labels**. The sizes of text for the labels of x and y axes, titles of x and y axes, and the figure title can be modified using the arguments *cex.axis*, *cex.lab*, and *cex.main*, respectively (Figure of this script is not shown because the font sizes are too large, but the values of *cex.axis*, *cex.lab* and *cex.main* are defined the same way as in Figure 23 below):boxplot(esc2xhigh2, notch = T, main = “Differential expression of transcription factors between fibroblasts and ESCs”, xlab = “Sample names”, ylab = “Log2-transformed read counts”, col.lab = “darkorange”, col.main = “red”, col.axis = “magenta”, **cex.lab = 2, cex.main = 2, cex.axis = 2**)  # After the font sizes are increased, both the title and the y axis annotation become too large to stay within the figure area. Two approaches can address this issue (see the next script and the subsection for changing the figure margin below).

To accommodate long title in multiple lines using the *paste()* function and the new line operator \n:boxplot(esc2xhigh2, notch = T, xlab = “Sample names”, ylab = “Log2-transformed read counts”, col.lab = “darkorange”, col.main = “red”, col.axis = “magenta”, cex.lab = 2, cex.main = 2, cex.axis = 2, main = **paste(“Differential expression of transcription factors”, “\nbetween fibroblasts and ESCs”))**  # Please note that the slash in the new line operator is a backslash, not a forward slash (Figure of this script is not shown, but the split title is the same as seen in Figure 23 below. In fact, Figure 23 uses this very script but its margin was pre-defined using the *par()* fuction and the *mar* argument as described below.).

## 21. Manipulate Figure Field

You can remove the frame of your graph using the logical argument *frame = FALSE*:boxplot(esc2xhigh2, notch = T, col = c(rep(“lightblue”, 4), rep(“bisque”, 3)), border = c(rep(“red”, 4), rep(“violet”, 3)), **frame = F**) # Please note that if you use the argument *axes = F*, you will remove both the frame and the axes (see below).

Now, you see the graph without the frame but the axes remain (Figure 21 without the horizontal lines, which are added using the next script below).

**Add a grid across the figure field.** You can add a grid to a figure. We just add horizontal lines since the vertical lines will overlap with the whiskers. To do this, we use the function of *abline()*. This must be done in two steps. Since you just run the above script, we add the horizontal lines subsequently using the function of *abline()*, which adds straight lines across the boxplot you just generated (Figure 21):**abline(h = c(0, 5, 10, 15, 20), col = “lightgray”, lwd = 2, lty = 2)**  # The numeric vector c(0, 5, 10, 15, 20) specifies the locations of the lines added; The argument *col* specifies the color of the line. The list of available colors can be found using the command of *colors()*. *lwd* specifies the thickness of the lines added relative to the default, which is 1; The argument *lty* specifies the line type in R graphics and the options are: 0, blank; 1, solid; 2, dashed; 3, dotted; 4, dotdash; 5, longdash; 6, twodash.

**Add background color to a figure**. You can add background color to your figure by using the parameter function *par()* and the background argument *bg*:**par(bg = “yellow”)**boxplot(esc2xhigh2, notch = T, col = c(rep(“lightblue”, 4), rep(“bisque”, 3)), border = c(rep(“red”, 4), rep(“violet”, 3)), frame = F)

You can see that the background of the boxplots becomes yellow (Figure 22). Please note that once you set up the background it will be the background for any graph you generate subsequently. If you do not want this background anymore, you can cancel it using the *dev.off()* function.

dev.off()–# After you run this command, a message of “null device” appears, indicating that you will no longer have the yellow background in the subsequent figures.

## 22. Modify Margins, Figure Border, and Axes

**Change the margins of a plot**. When you increase the font sizes for the axis labels and titles, the text may be too large so that the margin cannot accommodate them as you have seen by running the R script above. One solution is to increase the margin. We can change the margin of a plot with the parameter function *par()* and the margin argument function *mar* = c(bottom, left, top, right). The default margins are *mar = c(5.1, 4.1, 4.1, 2.1).* Let us increase both the left and top margins to 5.1:**par(mar = c(5.1, 5.1, 5.1, 2.1))**

Now, run the following boxplotting again:boxplot(esc2xhigh2, notch = T, main = paste(“Differential expression of transcription factors”, “\nbetween fibroblasts and ESCs”), xlab = “Sample names”, ylab = “Log2-transformed read counts”, col.lab = “darkorange”, col.main = “red”, col.axis = “magenta”, cex.lab = 2, cex.main = 2, cex.axis = 2) # This script is essentially the same as the one describing the *paste()* function above. The only difference is that the figure margin is pre-defined rather than the default one. Now, the y axis annotation is visible (Figure 23).

Please note that all plots you will generate will use this set of margins. If you want to return to the default margins again, you can run *dev.off()* to terminate the specified parameters by *par()*.

**Change the limits of x and y axis**. The limit of the axes can be defined using *xlim* and *ylim*:boxplot(esc2xhigh2, notch = T, main = paste(“Differential expression of transcription factors”, “\nbetween fibroblasts and ESCs”), xlab = “Sample names”, ylab = “Log2-transformed read counts”, col.lab = “darkorange”, col.main = “red”, col.axis = “magenta”, cex.lab = 2, cex.main = 2, cex.axis = 2, **xlim = c(0, 9), ylim = c(−4, 20)**)

You can see that there is more space on all sides of the boxplots (Figure 24. Compare it with Figure 23).

**Add and modify axes**. The axes and figure frame can be modified using the function *axis()*. This can be achieved in two steps. First, you generate the boxplots without the axes and frame (Figure 25):boxplot(esc2xhigh2, notch = T, col = c(rep(“lightblue”, 4), rep(“bisque”, 3)), border = c(rep(“red”, 4), rep(“violet”, 3)), xlab = “Treatments and Cell Types”, ylab = “log2 read counts”, cex.lab = 2, col.lab = “blue”, lwd = 2, **axes = F**) # The argument *axes = FALSE* removes the axes, axis labels, as well as the frame. The argument *frame = FALSE* removes the frame of the plot only (see above). To see how these arguments work, you can change the value from F to T for each argument.

After plotting the boxplots, you can add the modified axes on any side using the function *axis()*. The following script adds and defines the x axis and its labels (Figure 26).

axis(side = 1, lwd = 3, lwd.ticks = 2, col = “blue”, labels = c(“F1”, “F2”, “F3”, “F4”, “E1”, “E2”, “E3”), at = c(1, 2, 3, 4, 5, 6, 7), col.ticks = “red”, cex.axis = 2, col.axis = “magenta”) # As you can see, this script adds the axis and the related features in the position of the traditional x axis. This script contains a lot of information. The argument *side* defines which side of the axis will be added, and the values are: 1, underneath the plot; 2, to the left of the plot; 3, above the plot; 4, to the right side of the plot. The argument *lwd* defines the thickness of the axis line. The default is 1. *lwd.ticks* defines the thickness of the ticks, and the default is *lwd.ticks = lwd*, which means *lwd.ticks* uses the value that *lwd* defines in the case *lwd.ticks* is not specified. However, when specifically defined, *lwd.ticks* will override *lwd* regarding the thickness of the ticks. The argument of *col* specifies the color of the axis line. The *labels* argument defines the labels of each sample or set of coordinates. Please note that you have to use a numeric vector to define the location of each label using the *at* argument. You can define the colors of the ticks using the argument *col.ticks,* which overrides *col*. The font sizes of the labels can be defined using *cex.axis*, and their colors defined using *col.axis*.

The following script adds and defines the y-axis (Figure 27)

axis(side = 2, lwd = 3, lwd.ticks = 2, col = “blue”, col.ticks = “red”, col.axis = “brown”, labels = c(0, 2.5, 5, 7.5, 10, 12.5, 15, 17.5), at = c(0, 2.5, 5, 7.5, 10, 12.5, 15, 17.5))  # This script places the y axis in the position of the traditional y axis. Please note that *axis()* allows you to add additional ticks and the corresponding coordinates using two vectors.

The following script adds and defines an x axis above the plot (Figure 28).


axis(**side = 3**, lwd = 3, lwd.ticks = 2, col = “blue”, labels = c(“F1”, “F2”, “F3”, “F4”, “E1”, “E2”, “E3”), at = c(1, 2, 3, 4, 5, 6, 7), col.ticks = “red”, cex.axis = 2, col.axis = “magenta”)


The following script adds and defines a y axis on the right side (Figure 29). This additional y axis will be helpful if there are many boxes.


axis(**side = 4**, lwd = 3, lwd.ticks = 2, col = “blue”, col.ticks = “red”, col.axis = “brown”, labels = c(0, 2.5, 5, 7.5, 10, 12.5, 15,17.5), at = c(0, 2.5, 5, 7.5, 10, 12.5, 15, 17.5))


## 23. Save Your Box Plot Onto Your Computer

Finally, you want to save your graphs on your computer. This can be done in three steps using the graphic saving device:(1).Open the *tiff* graphics devicetiff(“boxplot.tiff”)(2).Create the plotboxplot(esc2xhigh2, notch = TRUE, col = c(rep(“lightblue”, 4), rep(“bisque”, 4)), border = c(rep(“red”, 4), rep(“violet”, 3)), names = c(“F1”, “F2”, “F3”, “F4”, “ESC1”, “ESC2”, “ESC3”), boxwex = 0.3, staplewex =1, at = c(1:4, 6:8), lwd = 2.5, cex.axis = 1.5, col.axis = “darkgreen”, las = 3, xlim = c(1, 8), ylim =c (−1.6, 17), frame = F)(3).Close the devicedev.off() # A message of “Null device” appears on the screen.

Then, you will see your graph in the current working directory on your computer hard drive. You can save the boxplots in other formats with the similar three steps (jpeg, png, or pdf formats).

The above process is not convenient. In RStudio, you can save the graphs easily using the pulldown menu as in Windows. In the Plots pane, click Export and you will see a pulldown menu. Click “Save as Image”, in the resulting Window, select the file format, directory, image size, and give a file name, and then click Save to save your graph onto your computer hard drive.

## 24. Save and Reuse the Script

**Save the boxplotting code**. You may want to save the script as a file and use it on a later day so that you do not need to type the code again. The following is the procedure:

Highlight the script in the console using the mouse as you do in Word → Click the Edit tab in the RStudio main menu bar to bring about the pulldown menu → Click Copy → Click the File tab to bring about the File pulldown menu → Put your mouse curser over New File to bring about the New File pulldown menu → Click R Script → In the script window, paste your script using the Edit sub-menu → You can see your script in the script window now, click the File tab and choose “Save as” → In the popup dialogue window, choose your target folder and save the code using an informative name.

**Load and re-use the saved boxplotting code**. When you want to use the saved script for similar boxplotting, you can follow the procedure below:

From the File tab, click Open File and this will bring about a dialogue window. From this Window, find the folder and choose the R code file you have saved before → Click Open, and you will see that your script appears in the script window in the top left quadrant → Highlight the script, and copy it → Go to the Console window by clicking anywhere in this window → Paste your script onto the Console → As needed, edit the code. Use the mouse, and the left or right arrow keys, to navigate within the script → Run the code with your data of interest.

## 25. Generating Violin Plots Using the Package of *vioplot*

An enhanced version of boxplots is violin plots. Violin plots have all the components of boxplots, but they add information on data density along the data points in the form of smoothed histograms (also called density trace) along the data points.

There are many different software packages to prepare violin plots. Here, I briefly introduce the R package *vioplot* to demonstrate how to make violin plots using the same demo data as above. The package *vioplot* depends on the packages *sm*, *tcltk*, and *zoo* to function. Make sure these packages are installed and loaded. If you install *vioplot* without prior installation of *sm* and *zoo*, it will install the dependents *sm* and zoo automatically, but you may have to individually load all the packages before you will use *vioplot()*.

install.packages(“sm”) # This command install the R package *sm* onto your RStudio.install.packages(“zoo”) # This script install the *zoo* package onto your RStudio.install.packages(“vioplot”) # This script install the *vioplot* pacakge onto your RStudio.library(vioplot)  # Loading *vioplot* usually loads the dependents *sm* and *zoo* as well. To load a package, you can also click the box next to the package names in the Packages pane. Sometimes, you may not simultaneously open the dependents if you click the boxes next to the packages to load the packages.

Because of space limit, only one script is given to generate violin boxes using the function of *vioplot()* for the data matrix of *esc2xhigh2* (Figure 30). Readers are encouraged to explore the *vioplot* package by studying the package information [19].

vioplot(esc2xhigh2, col = c(rep(“lightgrey”, 4), rep(“bisque”, 3)), border = c(rep(“green”, 4), rep(“brown”, 3)), lwd = 3) # You will see the violin plots after running this script (Figure 30).

## Figures and Tables

**Figure 1 mps-03-00064-f001:**
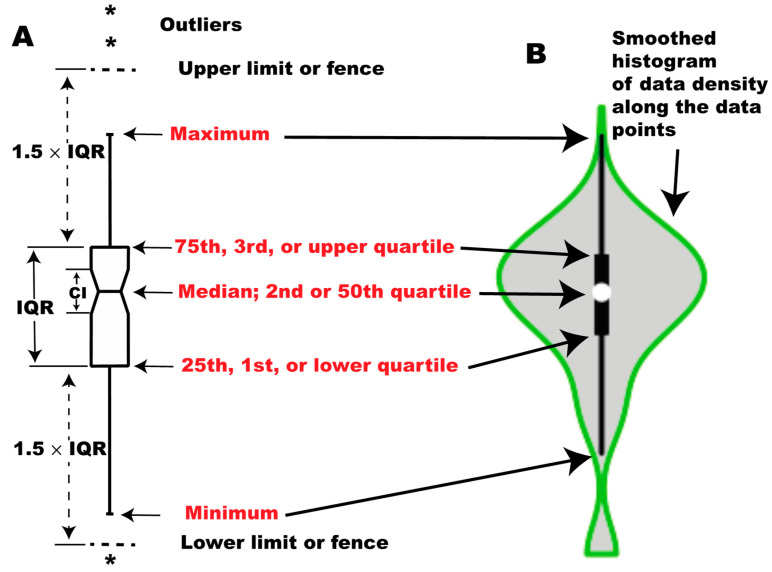
Configuration of boxplots. (**A**) conventional box plot; (**B**) violin plot.

**Figure 2 mps-03-00064-f002:**
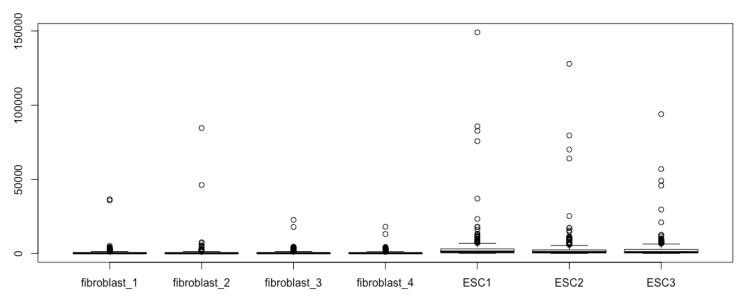
Boxplots prepared using the raw RNA-seq data, which cannot reveal the differential expressions.

**Figure 3 mps-03-00064-f003:**
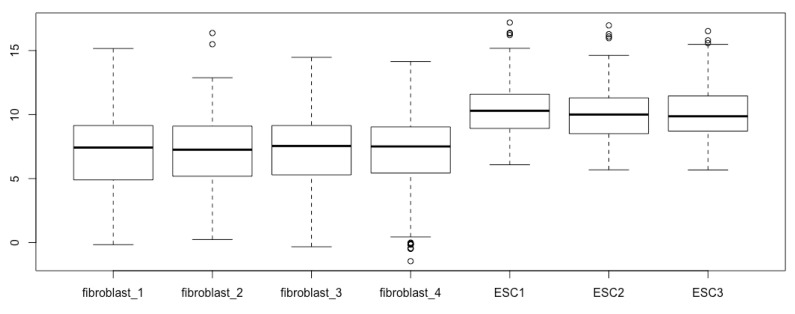
Prepare boxplots using the log2-transformed read counts for enhanced resolution.

**Figure 4 mps-03-00064-f004:**
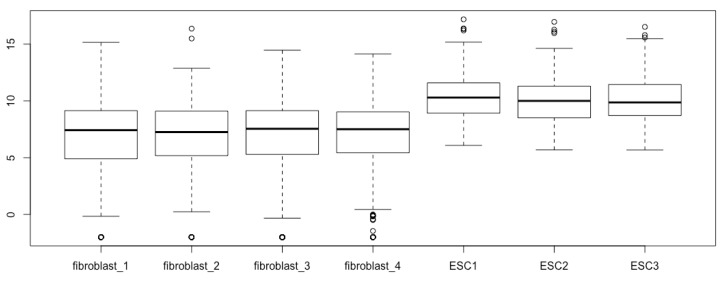
Visualize the -Inf outliers (open circles; compare with Figure 3).

**Figure 5 mps-03-00064-f005:**
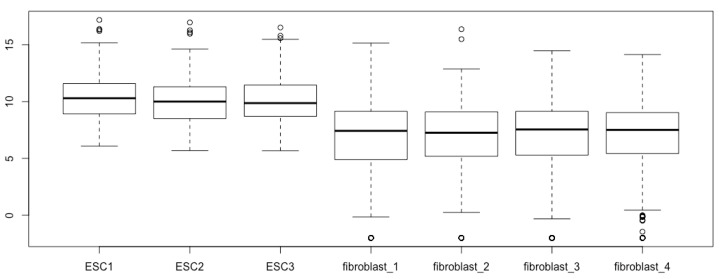
Re-order boxplots (compare it with Figure 4).

**Figure 6 mps-03-00064-f006:**
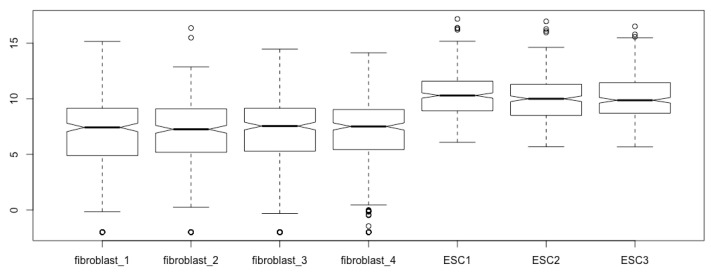
Add notches to boxplots using the argument notch = TRUE.

**Figure 7 mps-03-00064-f007:**
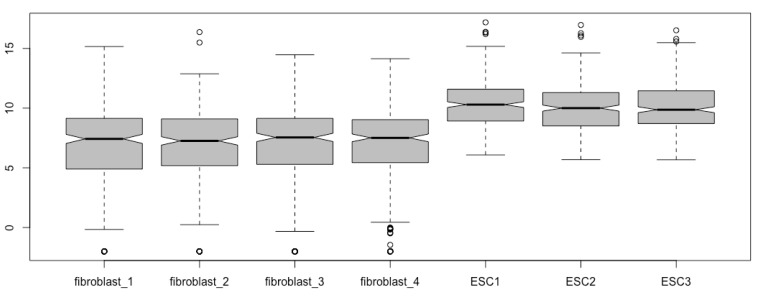
Fill the box bodies with a color using the argument *col = “gray”*.

**Figure 8 mps-03-00064-f008:**
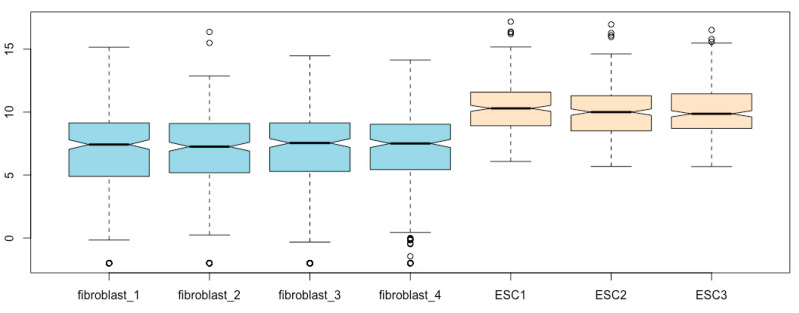
Fill box bodies with different colors for different cell types.

**Figure 9 mps-03-00064-f009:**
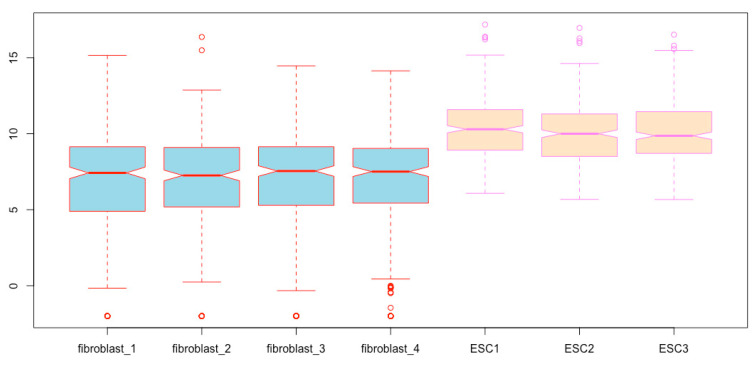
Assign different colors to all other boxplot components except for the box bodies using the argument of *border*.

**Figure 10 mps-03-00064-f010:**
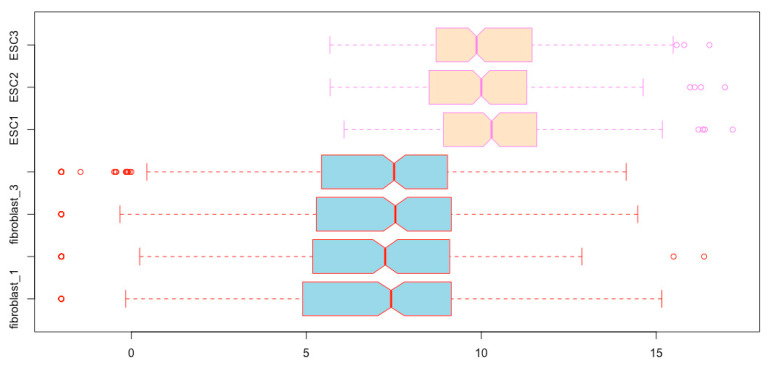
Place boxplots horizontally using the argument *horizontal = TRUE* or *T*.

**Figure 11 mps-03-00064-f011:**
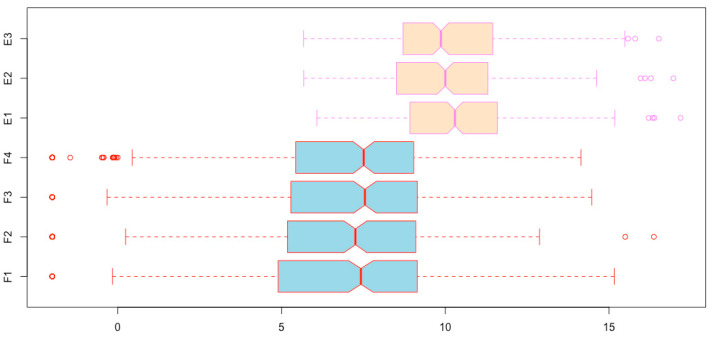
Assign shorter sample names for the horizontal boxplots using the R function *names()*.

**Figure 12 mps-03-00064-f012:**
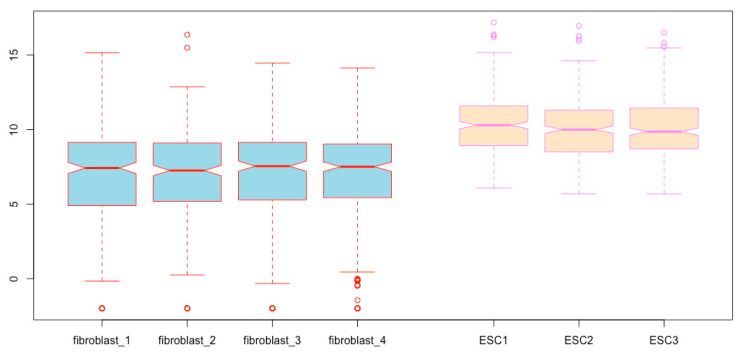
Spatially separate boxplots of different cell types.

**Figure 13 mps-03-00064-f013:**
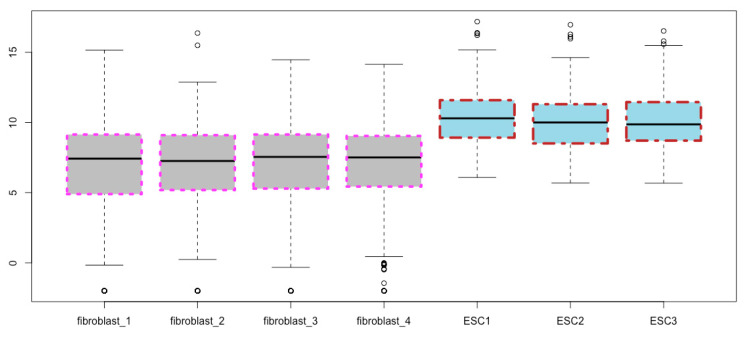
Define the line style, line thickness, and line colors of the box part of boxplots.

**Figure 14 mps-03-00064-f014:**
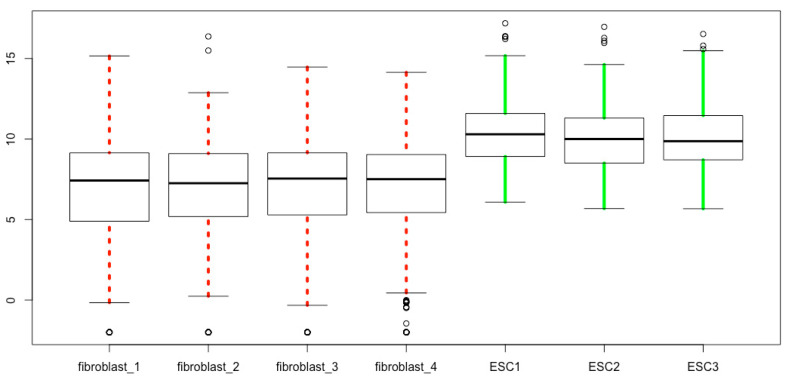
Define the line style, line thickness, and line color of the whisker parts of boxplots.

**Figure 15 mps-03-00064-f015:**
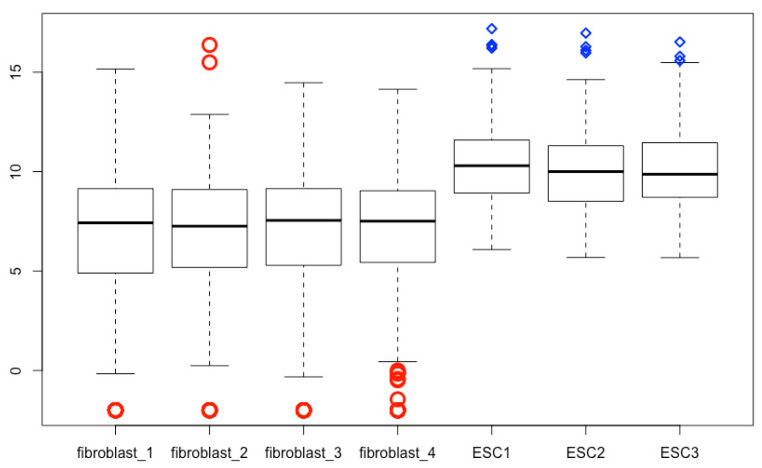
Define the point type, line thickness, point size, and point color of the outlier part of boxplots.

**Figure 16 mps-03-00064-f016:**
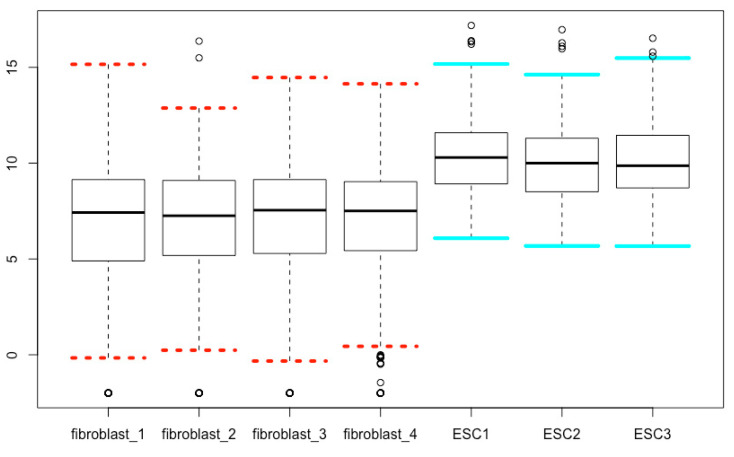
Define the line style, line thickness, line color, and line length of the staple part of boxplots.

**Figure 17 mps-03-00064-f017:**
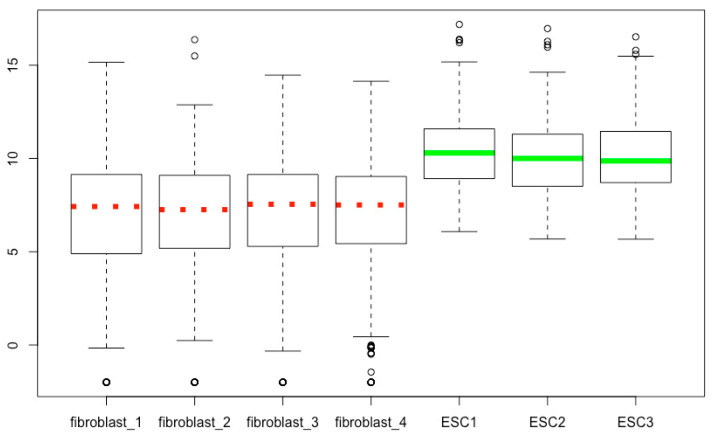
Define the line thickness, line style, and line color of the median lines of boxplots.

**Figure 18 mps-03-00064-f018:**
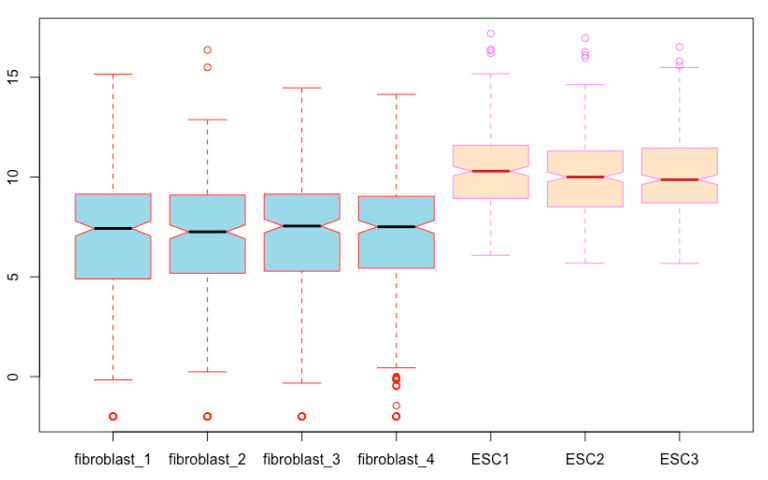
Modify some parts of boxplot by overriding features defined by the *border* function.

**Figure 19 mps-03-00064-f019:**
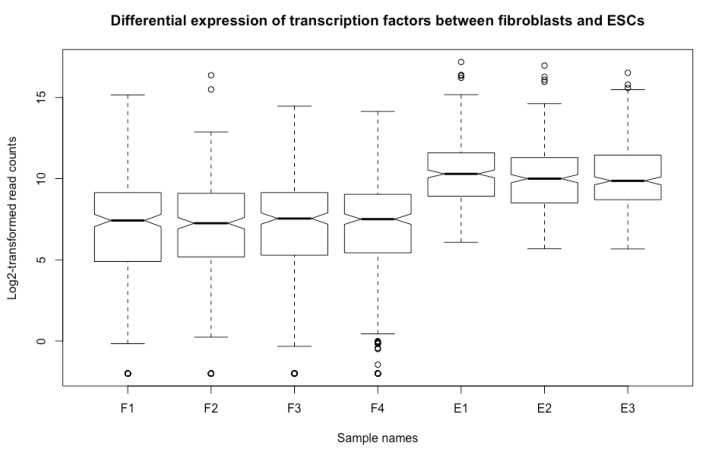
Add a figure title and annotation of axes.

**Figure 20 mps-03-00064-f020:**
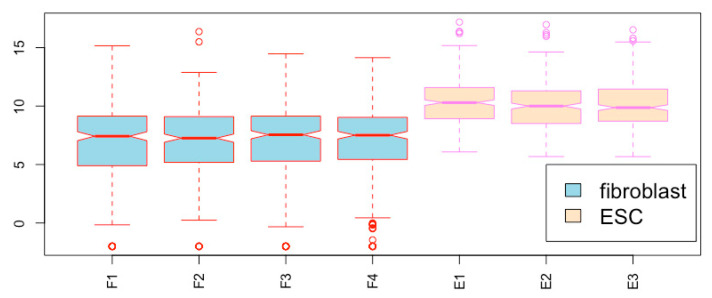
Define the orientation of axis labels, and add a legend to the figure.

**Figure 21 mps-03-00064-f021:**
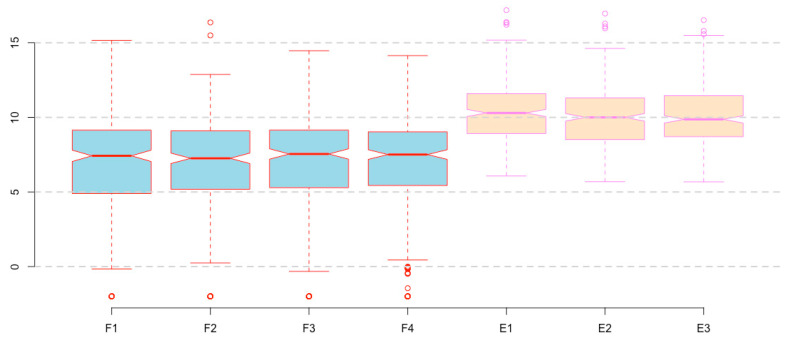
Remove the graph frame; add a grid to a figure.

**Figure 22 mps-03-00064-f022:**
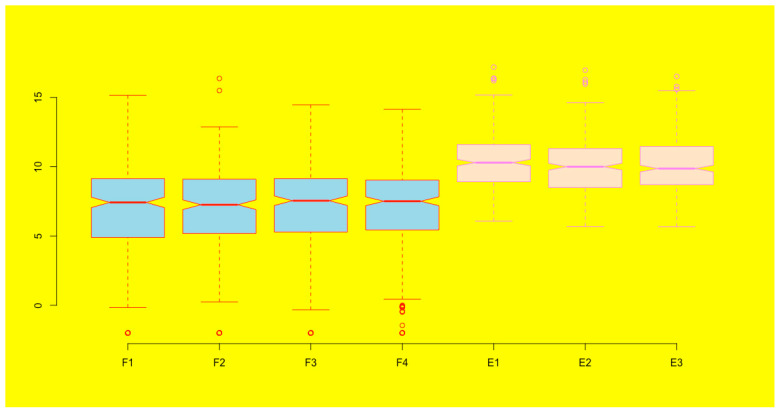
Add background color to a figure.

**Figure 23 mps-03-00064-f023:**
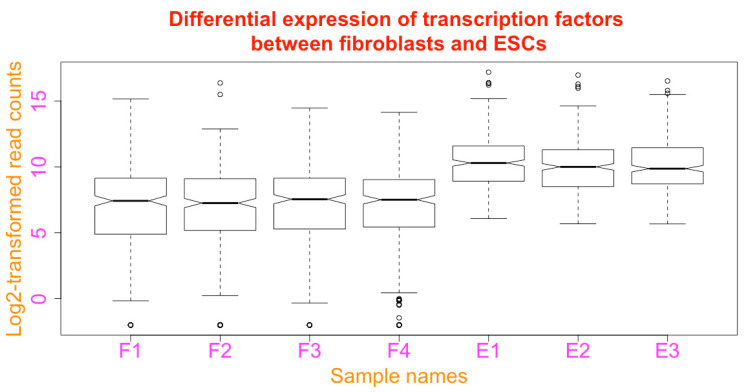
Change figure margins; add and modify figure title, axis titles and labels.

**Figure 24 mps-03-00064-f024:**
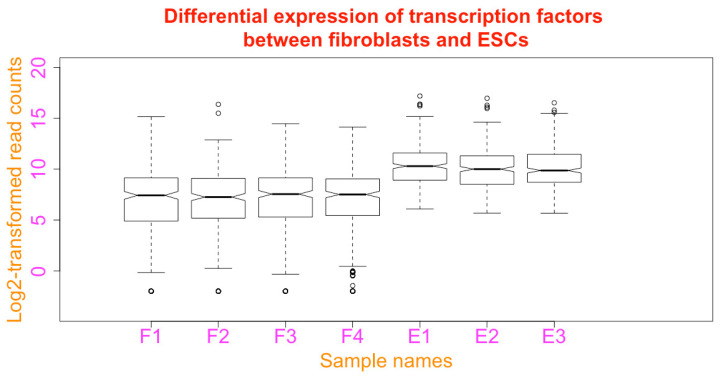
Change the limits of axes using the functions *xlim* and *ylim*.

**Figure 25 mps-03-00064-f025:**
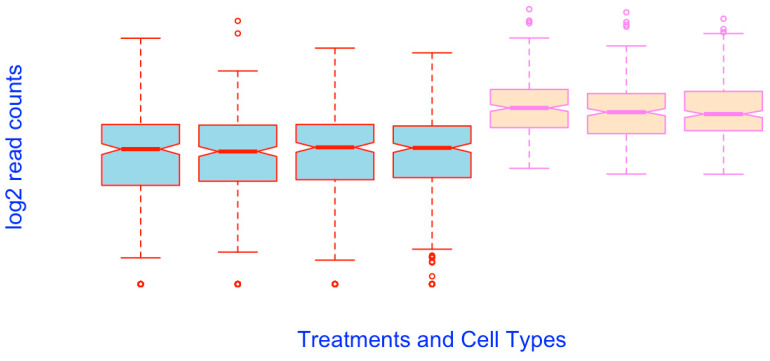
Remove the figure frame and axes using logical values.

**Figure 26 mps-03-00064-f026:**
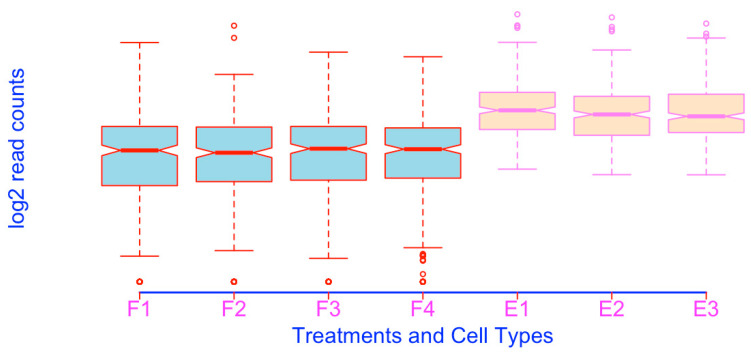
Add customized x-axis to a figure.

**Figure 27 mps-03-00064-f027:**
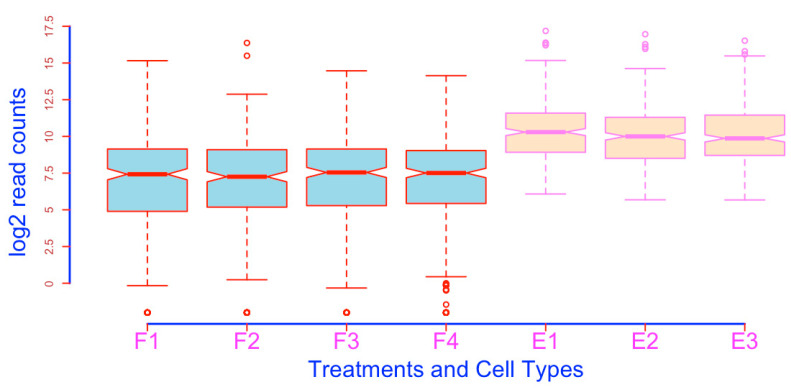
Add a customized y-axis to a figure.

**Figure 28 mps-03-00064-f028:**
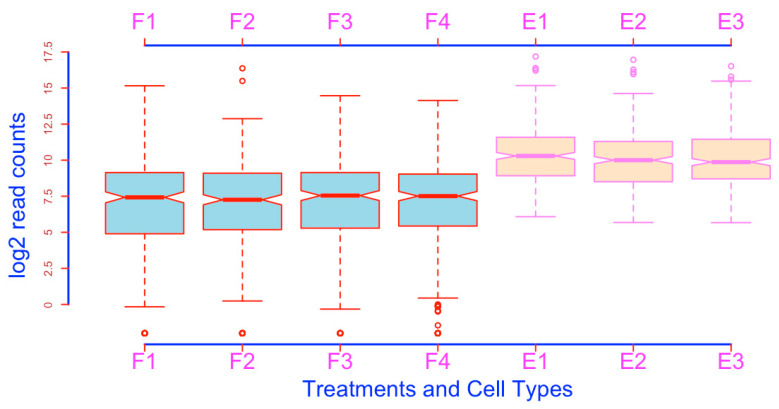
Add an x-axis above the boxplots.

**Figure 29 mps-03-00064-f029:**
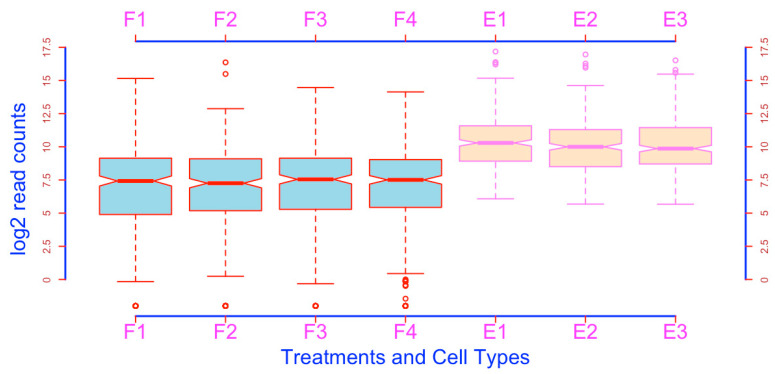
Add a y-axis to the right of boxplots.

**Figure 30 mps-03-00064-f030:**
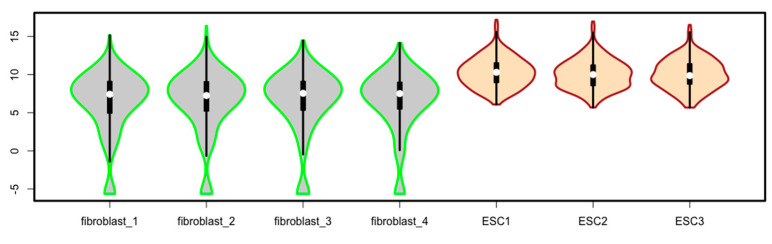
Violin plots generated using the R package of *vioplot*.

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
