# Peer review of "Become Competent within One Day in Generating Boxplots and Violin Plots for a Novice without Prior R Experience"

_mps, 2020, doi:10.3390/mps3040064_

Round 1

Reviewer 1 Report

This a well described protocol on how to produce boxplots in R, which will be extremely useful for people with none or limited knowledge of R. I only have a few minor comments.

  • I think the author could also quickly introduce somewhere the violin plots, which have the advantage over boxplots of showing the probability density of the data. This can be an important piece of information, especially in gene expression analysis.

  • It could also be interesting to cite an Introduction/protocol or introduce how to perform the statistical tests as it is one of the important steps in using boxplots as a way to show data. Typically, if you just compare Control vs Treatment without any selection for differentially expressed genes, do you have a significant difference in the global expression level?

Minor comments:

Line 77: soft wares -> software

Line 157: princt(x) -> print(x)

Line 313: “this tutorial provides techniques more than needed” -> this tutorial provides more techniques than needed

Line 453: “esc2xhigh_re-ordered” gives an error message: “Error in esc2xhigh_re - ordered <- esc2xhigh1[, c(5, 6, 7, 1, 2, 3, 4)] : object 'esc2xhigh_re' not found”. “esc2xhigh_re_ordered” should be preferred as a name.

For the “Deal with the -Inf in your matrix to be boxplotted” part: I don’t see the point of changing -Inf to 20 then to -2, I think the author can make a straight explanation with -2 rather than using 20 as a first example, which does not make much sense since we are dealing with a log2(0).

Line 643: “boxex” -> boxes

Line 667: “boxplot(esc2xhigh2, notch = TRUE, col=c(rep("lightblue", 4), rep("bisque", 4)), border” -> it should be rep("bisque", 3), it works with 4 but for consistency.

Line 686: “Chang sample names of boxplots.” -> Change.

Line 988: “and the default is lwd.ticks =” -> The default value is missing for lwd.ticks.

Author Response

Dear Reviewers,

I appreciate your time and effort. Your inputs make this tutorial better. I have revised the tutorial based on your comments. Please note that all revisions are highlighted in red.

Thank you for your further help.

Responses to Reviewer 1.

Q: “I think the author could also quickly introduce somewhere the violin plots, which have the advantage over boxplots of showing the probability density of the data. This can be an important piece of information, especially in gene expression analysis”.

Answer: A short introduction into violin plots is added and an illustration is added as Figure 1B. A separate R package vioplot is introduced and one violin plot is added as figure 30. Detailed explanation and illustration are not necessary and not possible due to the space limit. There are 30 figures already.

Q: “It could also be interesting to cite an Introduction/protocol or introduce how to perform the statistical tests as it is one of the important steps in using boxplots as a way to show data”. 

Answer: This is calculated by the software, and is stated in the text already.

Q: “Line 77, “Soft wares” should be software”.

Answer: Corrected.

Q: “Line 157, princt(x) shoud be print(x)”

Answer: Corrected.

Q: “Line 313: “this tutorial provides techniques more than needed” -> this tutorial provides more techniques than needed.”

Answer: Corrected.

Q: “Line 453: “esc2xhigh_re-ordered” gives an error message: “Error in esc2xhigh_re - ordered <- esc2xhigh1[, c(5, 6, 7, 1, 2, 3, 4)] : object 'esc2xhigh_re' not found”. “esc2xhigh_re_ordered” should be preferred as a name.”

Answer: This is a typo. Corrected. “re-ordered” should be “re_ordered” . In R, object Names should contain only letters, numbers, underscore characters (_), and dots (.). Although you can force R to accept other characters in names, you shouldn’t, because these characters often have a special meaning in R. Now, I add a note after this corrected script: “Please note that R object names use dot, underscore, number and letters only, and hyphen is not allowed. Please try esc2xhigh_re-ordered <- esc2xhigh1[, c(5, 6, 7, 1,2,3, 4)], and you will find an error message.”

Q: “For the “Deal with the -Inf in your matrix to be boxplotted” part: I don’t see the point of changing -Inf to 20 then to -2, I think the author can make a straight explanation with -2 rather than using 20 as a first example, which does not make much sense since we are dealing with a log2(0).”

Answer: There are other ways to treat log2(0). For example, replacing 0 with a very small number, say 0.01, before log2 transformation. Here, I offer one solution. The reason to replace -Inf with 20 is that it is difficult to find out the smallest values in the matrix because of the -Inf. We can temporally replace -Inf with a very large value not present in the matrix (it could be any positive number greater than the largest value in the matrix so that we can easily find out the smallest value in the matrix. Finally, we replace 20 with a value smaller than the smallest value in the matrix.

Q: “Line 643: “boxex” -> boxes”

Answer: Corrected.

Q: “Line 667: “boxplot(esc2xhigh2, notch = TRUE, col=c(rep("lightblue", 4), rep("bisque", 4)), border” -> it should be rep("bisque", 3), it works with 4 but for consistency.”

Answer: This is a typo in this line and in other places. I am glad the meticulous reviewers pick up this typo. Corrected in this line and other places as indicated by red highlight.

Q: “Line 686: “Chang sample names of boxplots.” -> Change.”

Answer: Corrected.  I revise a little bit though. The new subsection title is now: “Changing sample names of boxplots”.

Q: “Line 988: “and the default is lwd.ticks =” -> The default value is missing for lwd.ticks.”

Answer: In fact, the default value is not missing, and it is lwd. I admit that This original sentence is confusing. To make it clear, I have revised the sentence as: “lwd.ticks defines the thickness of the ticks, and the default is lwd.ticks = lwd, which means lwd.ticks uses the value that lwd defines in the case lwd.ticks is not defined. However, when specifically defined lwd.ticks will override lwd” regarding the thickness of the ticks. The subclause of the sentence was added.

Reviewer 2 Report

The manuscript seems to be a good introduction for users not familiar with R to create boxplots. It gives an introduction to basic R commands and basic data editing functionalities. However, as such, the manuscript does not contain any scientific value, for which I vote for rejecting it.

Author Response

Responses to Reviewer 2.

Q: “The manuscript seems to be a good introduction for users not familiar with R to create boxplots. It gives an introduction to basic R commands and basic data editing functionalities. However, as such, the manuscript does not contain any scientific value.”

Answer: I respectfully disagree. I am also glad that the other two reviewers are positive. In fact, I had no plan to make this tutorial public at the time I prepared it for my lab and students. My friends love this tutorial when I privately shared it with individuals. I then realized that it will help millions of bench scientists. I do agree that the direct scientific value of this tutorial is limited, but the indirect scientific values are huge since millions of scientists/students will benefit from it. As many top journals including Nature Methods suggested, wider use of boxplots will increase the reproducibility of research findings (see literature cited in my introduction of this tutorial).  These benefits are discussed in my introduction.  In fact, Nature Protocols also publishes common and classic tutorial. For examples, Nature Protocols published tutorial for packaging lentiviral vectors, which thousands of labs were already using it. But, a standardized protocol for lentiviral packaging still helps many labs (See publication of “Production, concentration and titration of pseudotyped HIV-1-based lentiviral vectors. Robert H Kutner1, Xian-Yang Zhang1 & Jakob Reiser). Another example, “Current Protocols in Molecular Biology” publishes very basic and classic techniques, which have been served as the bible in the community of molecular biology. I agree with the reviewer 2 that this tutorial is “a good introduction for users not familiar with R to create boxplots”, and it will benefit the entire scientific community.

Please note that this tutorial will motivate millions of students/scientists without prior R experiences. The bench workers were intimidated by the R programming platform. In fact, I tried to learn R several times over several years. Each time, I started and after one to two weeks I quitted. I started gain and quitted again from beginning. I finally mastered R and can do a lot of different analyses after spending focused three months. My current tutorial makes an otherwise lengthy and difficult learning process easy and quick one.  Another point is that this tutorial will save hundreds of million professional hours (albeit other professions rather than statistics, data sceince, and computer science). I spent three months of PI times to master R to prepare heat maps and boxplots and others when I did not have access to such a tutorial. Now, one just need one day only to be proficient in preparing boxplots. This translates to billions of dollars if one professional hour cost $25.

Reviewer 3 Report

This is an informative and useful protocol for new users without prior R experiences to generate boxplots. Steps are clearly defined and easy to flow. This protocol is valuable to the research community and would be very helpful for new user because of its step-by-step guidance.

Specific Comments:

  1. One page 2, line 49, it indicated “(add the heat map chapter)”: is this to cite a chapter or figure? Please clarify.
  2. Page 3, line 88: “The data used in this tutorial is RNA-seq normalized read counts for 310 human transcriptional factors that are differentially expressed between hESCs and fibroblasts”.
  • Since the example used in this paper is about comparing genes in fibroblasts (differentiated cells, somatic cells) versus ESCs (pluripotent stem cells, naïve stem cells expressing all pluripotent markers), a little more background about these cells would be helpful.
  1. Page 15, “Add notch to boxplot”:
  • It’s a great addition to add Notch to the boxplot, however, if the sample size is small, it may show an error. This will be resolved when taking off the “add Notch”. Please include this information in this section for researchers who perform analysis of small sample size.

Author Response

Responses to Reviewer 3.

Q: “One page 2, line 49, it indicated “(add the heat map chapter)”: is this to cite a chapter or figure? Please clarify.”

Answer: This citation is a book chapter. It is in production stage, and will be published soon. Citation is added now.

Q: “Page 3, line 88: “The data used in this tutorial is RNA-seq normalized read counts for 310 human transcriptional factors that are differentially expressed between hESCs and fibroblasts”.

Since the example used in this paper is about comparing genes in fibroblasts (differentiated cells, somatic cells) versus ESCs (pluripotent stem cells, naïve stem cells expressing all pluripotent markers), a little more background about these cells would be helpful.”

Answer: More information is given, and the citation is provided although the manuscript is still under revision after the first round of review.

Q; “Page 15, “Add notch to boxplot”: It’s a great addition to add Notch to the boxplot, however, if the sample size is small, it may show an error. This will be resolved when taking off the “add Notch”. Please include this information in this section for researchers who perform analysis of small sample size.”

Answer: A note has been added after that script as suggested.